

# Intercomparison of stratospheric temperature profiles from a ground-based microwave radiometer with other techniques

Francisco Navas-Guzmán[1], Niklaus Kämpfer[1], Franziska Schranz[1], Wolfgang Steinbrecht[2], and Alexander Haefele[3]

[1]Institute of Applied Physics (IAP), University of Bern, Bern, Switzerland
[2]Meteorologisches Observatorium Hohenpeißenberg, Deutscher Wetterdienst, Hohenpeißenberg, Germany
[3]Federal Office of Meteorology and Climatology MeteoSwiss, Payerne, Switzerland

*Correspondence to:* Francisco Navas-Guzmán (francisco.navas@iap.unibe.ch)

**Abstract.**

In this work the stratospheric performance of a relatively new microwave temperature radiometer (TEMPERA) has been evaluated. With this goal almost three years of temperature measurements (January 2015 - September 2016) from TEMPERA radiometer were intercompared with the measurements from different techniques as radiosondes, MLS satellite and Rayleigh

lidar. This intercomparison campaign was carried out at the aerological station of MeteoSwiss at Payerne (Switzerland). In addition, the temperature profiles from TEMPERA were used to validate the temperature outputs from SD-WACCM model. The results showed in general a very good agreement between TEMPERA and the different instruments and the model with a high correlation (higher than 0.9) in the temperature evolution at different altitudes between TEMPERA and the different datasets. An annual pattern was observed in the stratospheric temperature with in general higher temperatures in summer than

in winter and with a higher variability during wintertime. A clear change in the tendency of the temperature deviations was detected in summer 2015 which was due to the repair of an attenuator in the TEMPERA spectrometer. The mean and the standard deviations of the temperature deviations between TEMPERA and the different measurements were calculated for two periods (before and after the reparation) in order to quantify the accuracy and precision of this radiometer along these almost three years. The results showed absolute biases and standard deviations lower than 2 K for most of the altitudes and

comparisons proved the good performance of TEMPERA to measure the temperature in the stratosphere.

## 1 Introduction

The thermal structure of the atmosphere is one of the most important atmospheric characteristics for determining chemical, dynamical and radiative processes in the atmosphere. In the stratosphere, temperature can influence chemical processes, and its vertical profile is fundamental for investigating other atmospheric species as for example ozone or water vapor (Haefele et al.,

2009; Stähli et al., 2013; Moreira et al., 2015). In addition, stratospheric temperature is also a very important indicator of climate change (Randel et al., 2009). The temperature trends can provide evidence of the roles of natural and anthropogenic climate change mechanisms. Several studies have shown a detectable observed pattern of tropospheric warming and lower stratospheric cooling during the last few decades of the twentieth century which is very likely related to anthropogenic emissions of trace





gases, ozone and aerosols (Ramaswamy and Schwarzkopf, 2002; Santer et al., 2006; Schwarzkopf and Ramaswamy, 2008; Randel et al., 2009; Bindoff et al., 2013).

Stratospheric temperatures can present a large variability along the time, specially during winter. For example, the stratosphere can experience sudden temperature increases (Sudden Stratosphere Warming, SSW) due to dynamical processes where

the temperature can change by several tens of degrees within a very short time (Flury et al., 2009; Scheiben et al., 2012). These fast changes require measurement techniques with high temporal and spatial resolution in order to be able to monitor these processes in the stratosphere.

The in situ technique of radiosonde is extensively used for tropospheric temperature measurements due to its high vertical resolution. However, in the stratosphere, they are only able to cover the lower part, reaching maximum altitudes of around 35

km. In addition, they present also an important disadvantage against other techniques and it is their low temporal resolution since in the best of the cases they are only launch four times a day.

At present, stratospheric temperature profiles are mostly obtained by means of remote sensing methods as lidars and microwave radiometers. Rayleigh lidars have been shown to be a powerful tool to monitor the temperature in the middle atmosphere with a high spatial and temporal resolution (Hauchecorne and Chanin, 1980; Keckhut et al., 2001; Steinbrecht et al.,

2009). However this technique's main drawback is that they can not be operated either daytime or under cloudy or rainy conditions. In this sense microwave radiometer measurements can overcome these difficulties, since the measurements in the microwave region are almost not affected by liquid water and the radiometers can be continuously operated providing temperature profiles with a reasonably good spatial and temporal resolution. Most of the microwave radiometers for stratospheric temperature measurements are operated on board of satelllites (e.g. MLS instrument on the Aura satellite as described in Wa-

ters et al. (2006), AMSU-A instrument on the Aqua satellite as described in Aumann et al. (2003) and SABER instrument on the TIMED satellite as described in Remsberg et al. (2003)).

The possibility of ground-based microwave radiometry for stratospheric temperature measurements was shown for the first time in Waters (1973) and it has been recently implemented (Shvetsov et al., 2010; Stähli et al., 2013). The technique is based on the stratospheric thermal emission from high-rotational, magnetic dipole transitions of molecular oxygen around 53 GHz.

Ground-based microwave radiometer measurements present as main advantages that they can provide unattended continuous measurements of temperature profiles in almost all weather conditions with a reasonably good spatial and temporal resolution in the altitude range between 20 and 50 km above see level (asl). In addition, long-term measurements in a fixed location allow the local atmospheric thermodynamics to be characterized. In this study we are going to present almost three years of stratospheric temperature measurements from the TEMPErature RAdiometer (TEMPERA) which has been designed and

built by the Institute of Applied Physics of the University of Bern (Switzerland). This is the first ground-based microwave radiometer that is able to retrieve temperature measurements in the troposphere and in the stratosphere at the same time. Tropospheric retrievals from this radiometer have been evaluated in detail in other studies (Stähli et al., 2013; Navas-Guzmán et al., 2014, 2016). In this work we will focus on the stratospheric performance of TEMPERA (from 20 to 50 km) comparing its measurements with the ones from different instruments and techniques as radiosondes, satellite and lidar measurements. In

addition TEMPERA profiles will be used to validate the temperature outputs from SD-WACCM model.





The results obtained in this study provide a detailed evaluation of the temperature retrievals from TEMPERA radiometer. The paper has been organized in the following way. The description of the different instrumentation used in this work is introduced in Section 2. Section 3 presents a detailed description of the methodology used for the microwave temperature retrievals. Section 4 presents the results of the different comparisons of RS, MLS satellite, lidar, SD-WACCM versus TEMPERA radiometer.
And finally, we conclude with a summary of the key findings in Sect. 5.

## 2 Experimental site and instrumentation

A special campaign has been set up at the aerological station in Payerne (46.82° N, 6.95° E; 491 m above sea level (asl), Switzerland) of the Swiss Federal Institute of Meteorology and Climatology (MeteoSwiss). For this campaign, the TEMPERA radiometer was moved from the ExWi building of the University of Bern (Bern, Switzerland) to Payerne in December 2013.
The main goal of this campaign is to assess the tropospheric and stratospheric performance of TEMPERA using the versatile instrumentation available at this MeteoSwiss station (Navas-Guzmán et al., 2016). Particulary, in this study we will focus on the intercomparison of the stratospheric temperature profiles from TEMPERA.

Next, we will introduce the ground-based microwave radiometer called TEMPERA and all the other instrumentation used in this study. As it was already mentioned, TEMPERA radiometer is the first ground-based microwave radiometer which is able
to measure temperature profiles in the troposphere and in the stratosphere simultaneously (Stähli et al., 2013; Navas-Guzmán et al., 2014, 2016). It measures the microwave emission of the molecular oxygen in the 51-57 GHz range. The instrument consists of a frontend to collect the microwave radiation and two backends for the spectral analysis (a filter bank and a Fast Fourier Transform spectrometer (FFT)). The incoming radiation is directed into a corrugated horn antenna using an off-axis parabolic mirror. The antenna is characterized by a Half Power Beam Width (HPBW) of 4°. The detected signal in the two
backends is calibrated by means of an ambient hot load in combination with a noise diode. The calibration of the noise diode is perfomed every month using a hot (ambient) and a cold (liquid nitrogen) load. Figure 1 (left) shows a picture of TEMPERA radiometer where its different components can be observed: mirror (1), microwave absorbers (hot(2) and cold (3) load), receiver (4) and styrofoam window (5). Figure 1 (right) shows the isolated room where TEMPERA is located at the aerological station of MeteoSwiss at Payerne (Switzerland).

The tropospheric measurements by TEMPERA are performed by means of a filter bank. It covers a total of 12 frequencies uniformly distributed on the wing of the 60 GHz oxygen emission complex. Since tropospheric temperature measurements are not the topic of this study more details about technical aspects of the filter bank and the measurement protocol for this mode can be found in Stähli et al. (2013) and Navas-Guzmán et al. (2016).

For stratospheric measurements a second backend is used. It consists of a digital FFT spectrometer (Acqiris AC240) which
measures the two pressure-broadened oxygen emission lines centered at 52.5424 and 53.0669 GHz. The bandwidth of this spectrometer is 960 MHz and has a resolution of 30.5 kHz. The receiver noise temperature $T_N$ is around 480 K. More technical details about the different components of the microwave receiver as the IQ-Mixer or the local oscilator (LO) can be found in




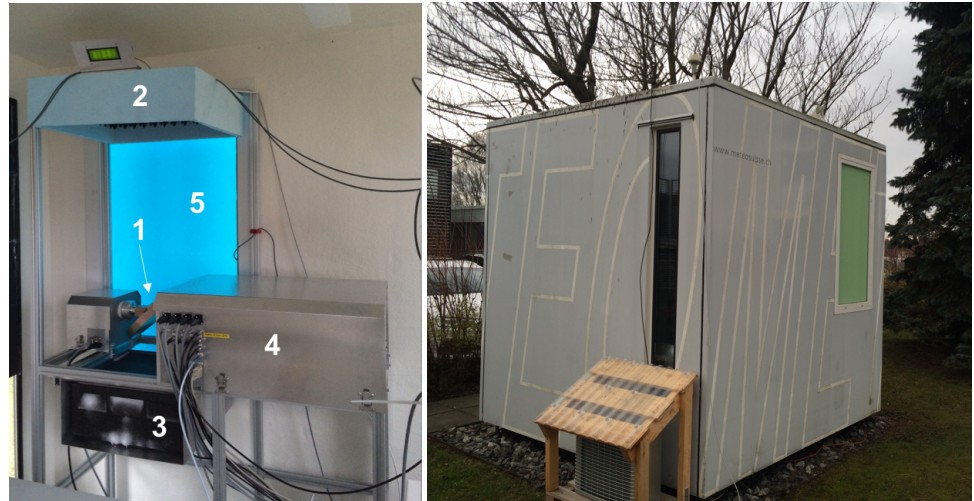

**Figure 1.** TEMPERA instrument at the MeteoSwiss Station in Payerne, Switzerland.

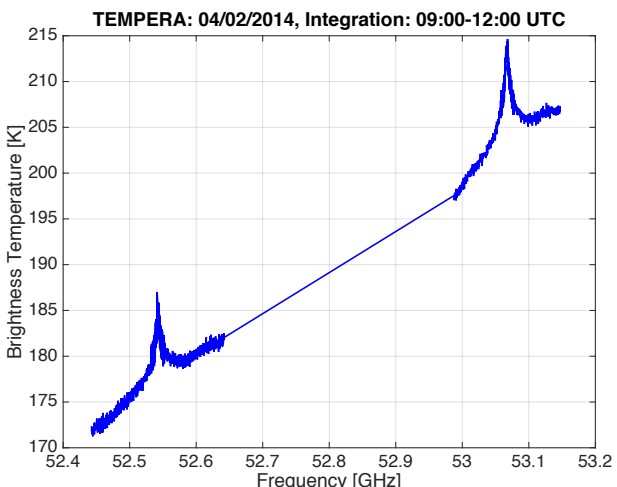

**Figure 2.** Spectrum of brightness temperatures measured with TEMPERA on 4 February 2014 from 09:00-12:00 (UTC). Only the FFT channels of the the first line at 52.5424 GHz and the second line at 53.0669 GHz used in the temperature retrievals are shown.

Stähli et al. (2013). An example of a calibrated spectrum (brightness temperature) measured with this spectrometer on 2 of February of 2014 is shown in Fig. 2.

A styrofoam window allows views of the atmosphere over different elevation angles (from $20°$ to $60°$). The operation of the instrument inside a laboratory presents as main advantage that the radiometer is protected against adverse weather conditions.




The frontend itself has additional temperature stabilization with Peltier elements in combination with a ventilation system leading to a stabilization of the frontend plate within $\pm 0.2$ K (Stähli et al., 2013).

Every measurement cycle takes 1 minute of duration and starts with a calibration using the hot load in combination with a noise diode for 9 s and followed by atmosphere measurements. These atmospheric measurements consist of a scanning from 20° to 60° elevation angles in steps of 5° (9 angles). The observations at all the angles are used for tropospheric measurements while only the observations at 60° elevation angle which take 15 seconds are used for stratospheric measurements (Stähli et al., 2013). Details about the methodology to obtain stratospheric temperature profiles from these measurements will be given in section 3.

Independent in-situ temperature measurement have been taken by means of radiosondes. They are regularly launched twice a day at the aerological station of Payerne since 1954. Radiosondes reach an altitude of 35 km, covering in this way only the lower stratosphere. The spatial resolution ranges between 10 to a maximum of 80 m with a highest resolution in the first seconds of the flight. The Swiss Radiosonde SRS-C34 introduced in 2011 uses a thermocouple for temperature measurements and a polymer hygristor for relative humidity measurements. Pressure is calculated from temperature and GPS altitude assuming hydrostatic equilibrium. The achieved uncertainties are $\pm$ 0.2 K for temperature, $\pm$ 2hPa (accuaracy increases with height) for pressure and $\pm$ 5 to 10% for relative humidity.

Stratospheric temperature have been also obtained from the Microwave Limb Sounder (MLS) instrument on board of the Aura satellite. MLS makes measurements of atmospheric composition, temperature, humidity and cloud ice in the upper troposphere, stratosphere and lower mesosphere since August 2004 (Waters et al., 2006). It observes thermal microwave emission from Earth's limb viewing forward along the Aura spacecraft flight direction, scanning its view from the ground to 90 km every 25 seconds. Aura is in a near-polar 705 km altitude orbit. As Earth rotates underneath it, the Aura orbit stays fixed relative to the sun; to give daily global coverage with 15 orbits per day. Aura is part of NASA's A-train group of Earth observing satellites. These satellites fly in formation with the different satellites making measurements within a short time of each other. Temperature profiles are retrieved from MLS measurements using radiances near the O2 spectral bands at 118 GHz for the stratosphere and mesosphere and at 239 GHz for the troposphere (Yan et al., 2016) using the optimal estimation theory (Rodgers, 2000). Four different versions of MLS data have been released to date. The initial version 1.5 (v1.5), was replaced by version 2.2/2.3 (v2) in 2007 and version 3.3/3.4 (v3) in 2010. The most recent production version, version 4.2 (v4), replaced v3 in February 2015. All the MLS data presented in this study correspond to the last version (v4).

Temperature measurements in the upper stratosphere have been also obtained from a lidar at Hohenpeißenberg, Germany (47.8° N, 11.0° E). This lidar has been operated since September 1987 by the German Weather Service (DWD) and has provided one of the longer NDACC time series (Steinbrecht et al., 2009). It emits intense ultraviolet light pulses at 353 nm generated from a Xenon Chloride excimer laser and a Hydrogen Raman cell. Light intensity scattered back from air molecules in the atmosphere (by Rayleigh scattering) is recorded as a function of altitude (=time from pulse emission to reception of backscattered light). Above the stratospheric aerosol layer, that is above 25 to 30 km, the returned light intensity is proportional to air density. Assuming hydrostatic equilibrium, this (relative) density profile can be integrated downward over altitude, providing a (relative) pressure profile. Division of the (relative) pressure profile by the (relative) density profile then yields the



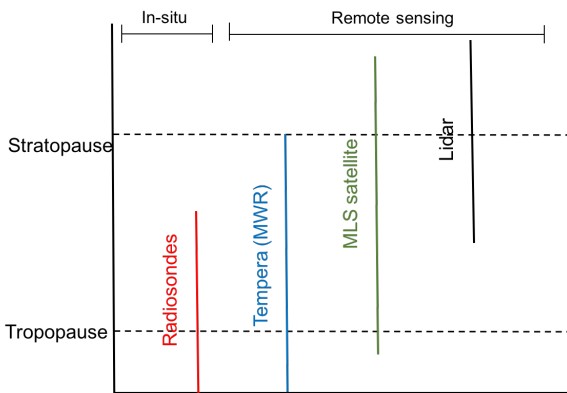

**Figure 3.** Measurement range for the different techniques used in this study (radiosondes, Tempera radiometer, MLS and lidar).

temperature profile. See Hauchecorne and Chanin (1980) for details. The method requires an initial guess for temperature (or pressure) at the far end around 70 to 80 km altitude, but because of the large increase of pressure with decreasing altitude, this choice of initial value has virtually no influence on the derived temperatures below 50 to 60 km altitude. The lidar requires clear nights for operation, and typically provides 80 to 90 nightly mean temperature profiles per year. Precision of the derived

temperature is about ±0.5 K at 30 km, ±1 K at 45 km, ±5 K at 60 km and ±10 K at 70 km (all 1 sigma). Vertical resolution is about 1.5 km. The lidar derived temperature has a low bias of about 2 K between 30 and 50 km, which is not well understood. See Steinbrecht et al. (2009) for details.

    Figure 3 shows the different ranges of measurements for each instrument used in this study (radiosondes, TEMPERA radiometer, MLS satellite and lidar). As we can see TEMPERA is the only instruments which is able to cover the full troposphere

and stratosphere.

## 3   Methodology

### 3.1   Temperature profiles from TEMPERA radiometer

Oxygen is a well-mixed gas whose fractional concentration is independent of altitude below approx. 80 km, so the microwave radiation contains information primarily on atmospheric temperature. The retrievals of stratospheric temperature profiles from

TEMPERA are based on the measurements of two oxygen emission lines centered at 52.54 and 53.06 GHz (see Fig. 2). The shape of these lines are governed by a pressure broadening mechanism up to 60 km of altitude, therefore the measured spectra can provide vertical information. The wings of the emission lines provide information of the radiation coming from low altitudes (higher broadening caused by higher pressure) while the center of the lines give information of the radiation coming from upper altitudes (smaller broadening and lower pressure). Both emission lines measured by TEMPERA are used at the

same time with a bandwidth of 200 MHz around the first line and of 160 MHz around the second line. Only measurements at




the observational elevation angle of 60° are taken for stratospheric measurements with the digital FFT spectrometer. It implies that during a measurement cycle the integration time with the FFT spectrometer is 15 s. In order to get a low enough noise level the measurements are integrated for half an hour which requires two hours of measurement time since only one quarter of the measurement time is spent for the digital FFT spectrometer (Stähli et al., 2013). Therefore the time resolution of the

stratospheric temperature profiles from TEMPERA radiometer is two hours.

Obtaining temperature profiles from the calibrated brightness temperature spectrum showed in Fig. 2 requires a solution of the radiative transfer equation. This is not unique and some statistical constraints are needed in order to obtain physically meaningful solutions. In our case we use the optimal estimation method (OEM) (Rodgers, 2000) by means of the radiative transfer model ARTS/QPack (Eriksson et al., 2011). The method is based on Bayes' probability theorem and detailed description about

it applied to TEMPERA measurements can be found in Stähli et al. (2013).

ARTS package implements the radiative transfer equation (forward model) to simulate the brightness temperature as:

$$\mathbf{y} = F(\mathbf{x}, \mathbf{b}) + \epsilon \tag{1}$$

where the vector $\mathbf{y}$ corresponds to the measured spectrum (brightness temperature), $\mathbf{x}$ is the true temperature profile, $\mathbf{b}$ contains some additional forward model parameters, and $\epsilon$ is the measurement noise.

The solution to the inverse problem is obtain using the Gauss-Newton iterative method, whose solution can be expressed in a matrix notation as follow:

$$\mathbf{x}_{i+1} = \mathbf{x}_i + \left( \mathbf{S}_a^{-1} + \mathbf{K}_i^T \mathbf{S}_\epsilon^{-1} \mathbf{K}_i^{-1} \right) \left[ \mathbf{K}_i^T \mathbf{S}_\epsilon^{-1} \left( \mathbf{y} - F(\mathbf{x}_i) \right) - \mathbf{S}_a^{-1} \left( \mathbf{x}_i - \mathbf{x}_a \right) \right] \tag{2}$$

where the vector $\mathbf{x}$ is the true temperature profile, $\mathbf{y}$ is the measured spectrum (brightness temperature), $\mathbf{x}_a$ is the a priori temperature profile, $\mathbf{S}_a$ is the a priori covariance matrix and $\mathbf{S}_\epsilon$ is the observation error-covariance matrix. The use of the

forward model is noted by $F$ and the vector $\mathbf{K}$ is the weighting function ($\mathbf{K} = \partial F / \partial \mathbf{x}$).

An important tool used very often in the OEM is the averaging kernel matrix $\mathbf{A}$ (Rodgers, 2000). This matrix describes the response of the retrieved temperature profile $\hat{\mathbf{x}}$ to the true temperature profile $\mathbf{x}$ and is defined as:

$$\mathbf{A} = \mathbf{D}_y \mathbf{K}_x = \frac{\partial \hat{\mathbf{x}}}{\partial \mathbf{x}} \tag{3}$$

where $\mathbf{K}_x$ is the weighting function matrix defined before and $\mathbf{D}_y = \partial F / \partial \mathbf{x}$ is the so called contribution function.

The rows of $\mathbf{A}$ are called the averaging kernels (AVK) and they describes the sensitivity of the retrieval for a certain height level to a perturbation at other levels. The sum of the AVK is called the measurement response (MR), which describes the contribution of measurement to the retrieved profile at a certain height.

The method needs an a priori temperature profile in order to constrain the solutions to physically meaningful results. As a priori profiles, monthly mean temperature profiles from radiosonde measuremens at Payerne from 1994 to 2011 are used in

the lower part (ground to 15 km) and mean MLS temperature profiles from a climatology are used in the upper part. As apriori covariance matrix $\mathbf{S}_a$ a function decreasing exponentially with a correlation of 3 km is used assuming a standard deviation of 2 K. For the observation errors the residuals of the inversion are considered (difference between the integrated spectra and the fit of the spectra). Under regular conditions these errors range between 0.5 and 1.5 K (Stähli et al., 2013).





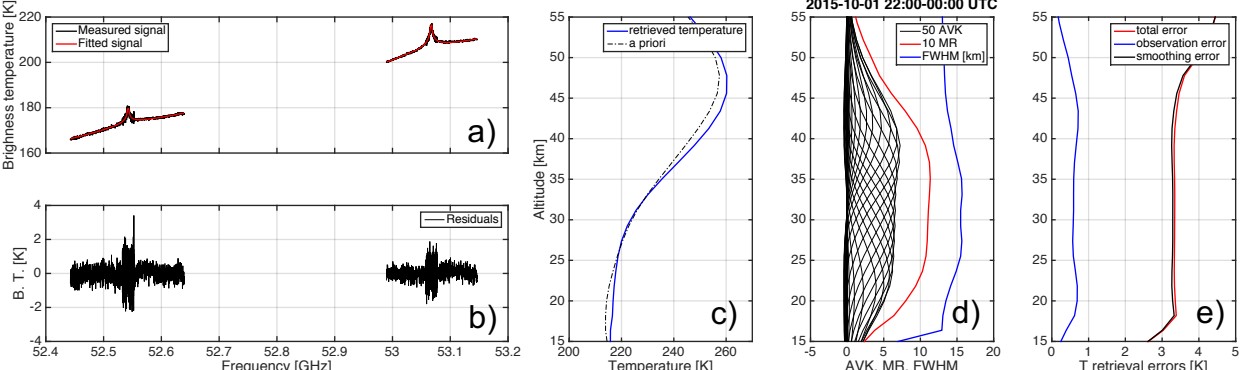

**Figure 4.** Temperature retrieval of 1 October 2015 using OEM. a) Brightness temperature measured with TEMPERA (black lines) compared with the forward model brightness temperature (red lines) obtained for this retrieval. b) Residuals for this inversion. c) Retrieved temperature and a priori profile. d) Averaging kernels, measurement response and FWHM [km] are plotted. d) Temperature retrieval errors.

In the radiative transfer calculations ($F(x,b)$) the absorption coefficients of the different species are calculated using different models: Rosenkranz (1998) for $H_2O$, Rosenkranz (1993) for $O_2$ and Liebe et al. (1993) for $N_2$. The density profiles of oxygen ($O_2$) and nitrogen ($N_2$) are incorporated by ARTS assuming standard atmospheric profiles for summer and winter (Anderson et al., 1986). In the case of tropospheric water vapour a profile with an exponential decrease is considered. This profile is

5 calculated with the measured surface water vapor density from a weather station and assuming a scale height of 2000 m (Bleisch et al., 2011).

Figure 4 shows an example of temperature inversion from TEMPERA measurements using OEM obtained on 1 October 2015 for the time interval from 22 to 00 UTC. In Fig. 4a we can observe that the forward model brightness temperatures (red lines) agree well with the measured brightness temperatures (black lines) excepting around the line center. The larger

differences observed in the center of the emission lines (see Fig. 4b) can be due to the Zeeman effect that is not incorporated in these retrievals (Stähli et al., 2013; Navas-Guzmán et al., 2015). Fig. 4c presents the a priori temperature profile used in the inversion (black dash line) and the retrieved temperature profile (blue line). Figure 4d shows the averaging kernels (black lines), the measurement response (red line) and the height resolution which is defined as the full-width at half-maximum (FWHM) of the averaging kernels (blue line). We can observe that for this inversion the height resolution ranges between 13 and 16 km.

The MR shows values larger than 0.8 in the range between 20 and 43 km, meaning that 80% of the contribution to the retrieved temperature profile comes from the measurements. These values decrease with altitude reaching 0.5 at 47 km for this case. We would like to point out that the altitude range of the stratospheric temperatures from TEMPERA radiometer used in this study correspond to levels with a high MR (higher than 0.8 in most of the alitutdes). Finally, the total, observational and smoothing errors are also calculated with this method and are shown in Fig. 4e.




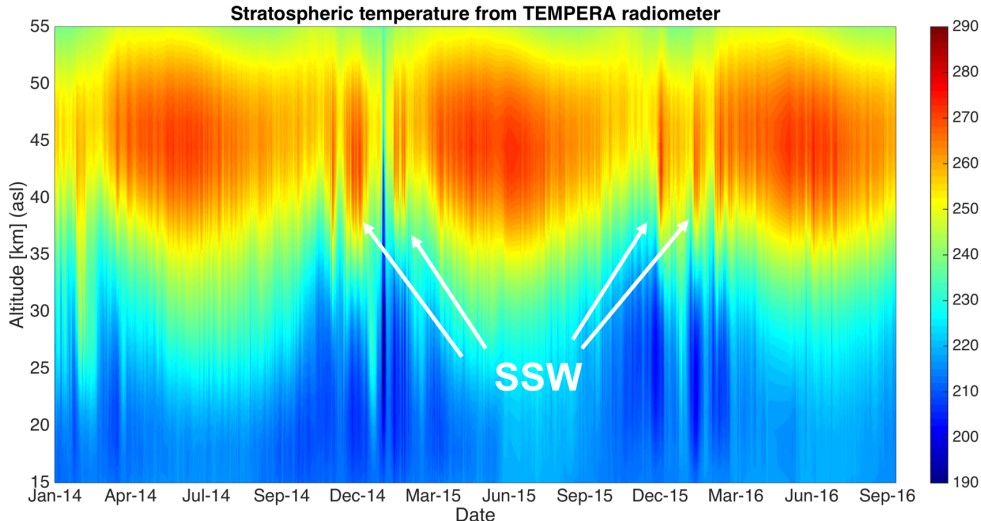

**Figure 5.** Stratospheric temperature evolution from TEMPERA radiometer. Some SSW events have been marked with white arrows.

In order to compare the temperature profiles from the different instruments (radiosondes, MLS satellite, lidar) and also from the WACCM model with the ones from TEMPERA radiometer the profiles are firstly interpolated to the pressure grid of TEMPERA and after that they are convolved using the averaging kernel of this radiometer in order to take into account the different height resolution. Equation 4 gives the expression to calculate the convolved temperature profiles:

$$\hat{\mathbf{x}}_r = \mathbf{x}_a + \mathbf{A}\left(\mathbf{x}_r - \mathbf{x}_a\right) \tag{4}$$

where $\mathbf{x}_a$ is the a priori profile of the radiometer, $\mathbf{A}$ is the averaging kernel and $\mathbf{x}_r$ is the interpolated reference profile.

## 4   Results: Evaluation of stratospheric temperature profiles from TEMPERA

TEMPERA radiometer has been almost continuously measuring since 2014 at the aerological station of MeteoSwiss at Payerne (Switzerland). Figure 5 shows the stratospheric temperature evolution obtained from TEMPERA for almost these three years

of measurements. From this plot a clear annual pattern can be observed with in general higher temperatures in spring and summer than in autumn and winter. Some interesting episodes can also be observed during the three presented winters, in which strong increases of temperature are measured for short periods in the upper stratosphere and could be identified as SSW. These increases of temperature in the upper stratosphere come many times associated with a decrease of temperature in the lower stratosphere being this pattern characteristic of SSW events. The measurements presented in the plot show the

importance of continuous observations for a fixed location, since the variability of atmospheric parameters such as temperature evinces the necessity of measurements with good temporal resolution.





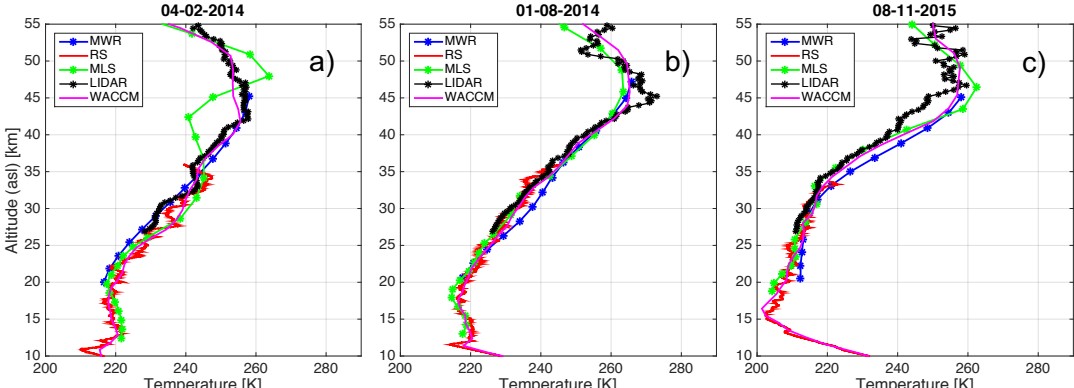

**Figure 6.** Stratospheric temperature profiles for night-time measurements from TEMPERA, RS, MLS, Lidar and WACCM model on (a) 4 February 2014, (b) 1 August 2014 and (c) 8 November 2015.

The temperature profiles from TEMPERA have been compared with the ones from other instruments and model, which have different spatial and temporal resolutions. Figure 6 presents three representative examples of stratospheric temperature profiles during winter, summer and autumn. Measurements from the different instruments and model (re)analyses show in general a good agreement in the range where they are comparable. Some evident differences are observed in the upper stratosphere

between MLS measurements and the other profiles on 4 February 2014. For the other two days is the lidar (black line) the one that presents deviations respect to the microwave measurements and the model in some ranges in the upper stratosphere. We would like to point out the good agreement observed between TEMPERA radiometer and most of the other techniques in these three cases. The examples also illustrate the different vertical ranges and the spatial resolutions from the different measurements. We can observe that radiosondes only cover the lower stratosphere but with a high spatial resolution, while

lidar measurements provide information in the upper stratosphere. MLS and TEMPERA are able to cover almost the whole stratosphere although their spatial resolution is lower.

In order to validate the accuracy and errors of the temperature profiles from TEMPERA radiometer a statistical analysis is performed with almost three years of measurements (January 2014 - September 2016). For this period, TEMPERA profiles are compared with the ones from very different techniques as they are RS, MLS satellite and lidar. In addition, profiles generated

by SD-WACCM model have been validated with the ones from TEMPERA. In the next sections the different comparisons are presented.

## 4.1 Comparison with RS

Stratospheric temperature profiles from TEMPERA have been compared with the ones from RS measurements for the period from January 2014 to September 2016. As it was indicated in previous sections RS are launched regularly twice a day (11

and 23 UTC) at the aerological station at Payerne since 1954. The TEMPERA profiles closest in time to the RS launch have been selected in order to do this comparison. A total of 1489 pairs of profiles are used in this statistics which were measured





under all weather conditions except rainy cases. The RS profiles were interpolated to the same altitude grid of TEMPERA radiometer and completed in the upper part (above 35 km, no RS meas.) with the measurement from TEMPERA in order to use the averaging kernels of TEMPERA in the convolution of the RS profiles.

Figure 7 shows the temporal evolution of the stratospheric temperature at different altitudes from TEMPERA and RS along almost these three years of measurements. The interpolated temperatures from RS have also been plotted (green lines) in order to visualise the smoothing effect on them when they are convolved with the averaging kernels of TEMPERA. In addition, the a priori temperature used for the TEMPERA inversions has also been displayed. The temperature deviations along this period between TEMPERA and RS are shown in the lower panels (black lines). We can observe in general a very good agreement between both instruments for the displayed altitudes with correlation coefficients higher than 0.9. An annual pattern is observed in the stratospheric temperature with higher temperatures in summertime than in wintertime. Again in this plot we can observe that the variability of the temperature is higher during winter than in other seasons, and some interesting events with a strong increase of the temperature have been detected (January 2014 and 2015, February 2016). The temperature deviations between TEMPERA and RS are in general small with most of the values below 3 K, although some short periods with larger discrepancies are also found (e.g. February 2015). We can also observe from these plots that the deviations at 27 km altitude are larger and noisier than for the other two altitudes. A remarkable feature observed in the temperature deviation lines at all the profiles is a small step in summer of 2015. This step is more evident in the two higher altitudes (27 km and 33 km) where the deviations changed from positive to negative. The effect is smaller at the lowest altitude (21.5 km) and it looks to have an opposite behaviour, changing from negative or almost zero deviations to positive deviations after the step happens. This change of tendency could be due to the fact that there was a repair of an attenuator of the FFT spectrometer in summer 2015. It seems that after this repair the brightness temperature spectra measured by the FFT were slightly affected and some small differences in the retrieved temperature are observed.

In order to take into account this instrumental modification and characterize possible changes in the accuracy and precision of TEMPERA radiometer the statistical analysis between TEMPERA and the other measurements (RS, MLS, lidar and WACCM) is carried out over two different measurement periods. From here on, period 1 will be referred as the period before the attenuator in the FFT spectrometer was changed (January 2014-June 2015) while period 2 will be referred as the period after this repair (July 2015-September 2016). In addition, a seasonal distinction has been performed to take into account the larger atmospheric variability that could be observed during wintertime. The atmospheric conditions during wintertime could produce larger deviations between the different measurements than due to the different measurement techniques. With this goal, the year has been split in two seasons, winter and summer. Winter measurements refer to those measurements taken between October and March while summer measurements refer to observations between April and September.

Figure 8 shows the mean and the standard deviations between TEMPERA and RS which have been calculated for all the measurements in each period (black lines) and also for winter and summer seasons of the different periods (blue and red lines, respectively). From this plot we can observe that there is a clear change in the mean bias between TEMPERA and RS for periods 1 and 2. The mean bias for period 1 ranged between -0.3 K at 20 km and 2.6 K at 28.5 km showing in general a positive deviation at most of altitudes. The mean bias in period 2 showed negative values for most of the altitudes with values ranging





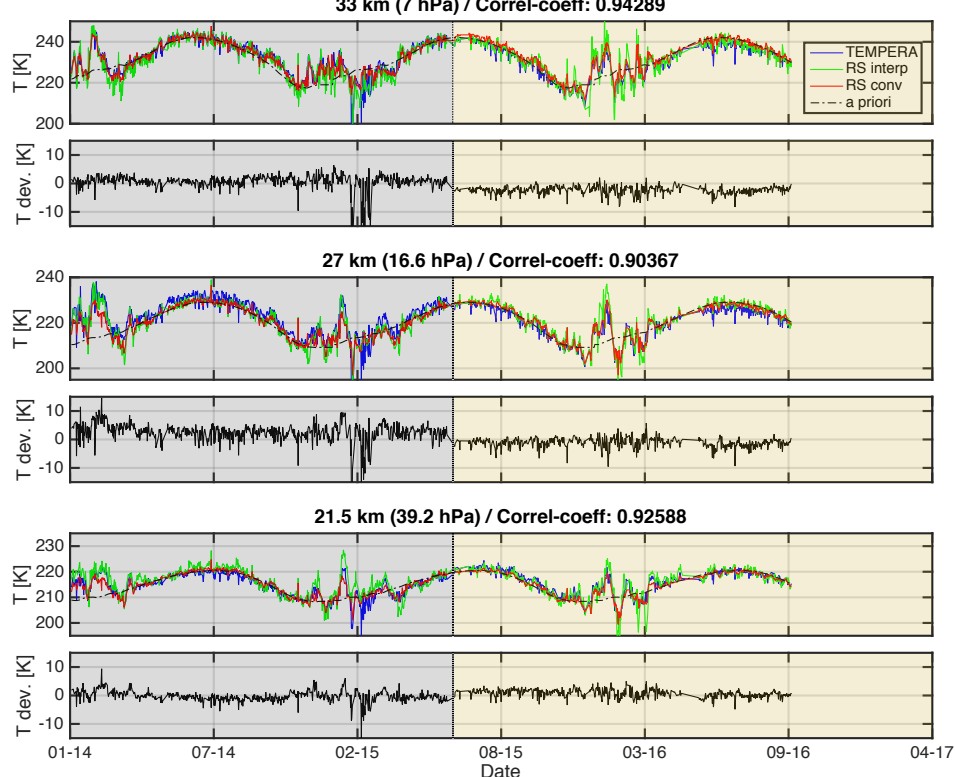

**Figure 7.** Stratospheric temperature evolution and temperature deviations at different altitudes for RS and TEMPERA. Different background colors are used to distinguish between period 1 and 2 (gray and light brown, respectively).

between 0.9 K (20 km) and -2.3 K (32 km). There is also a clear difference in the standard deviation observed for both periods. Period 1 showed much larger standard deviations than period 2 with values that range between 1.9 K (21 km) and 3.5 K (28.5 km). The standard deviations for period 2 were smaller and much more constant in height with values ranging between 1.3 K (34 km) and 1.7 K (26.5 km). These results show a change in the sign of the bias between TEMPERA and RS when the

5    attenuator of the FFT spectrometer was repaired in June of 2015, although in term of absolute values the differences were not very significant. However, concerning the standard deviations period 2 showed lower values than period 1 indicating a higher precision of TEMPERA radiometer after the repair respect to the reference RS measurements. If we have a look at the seasonal behaviour of the bias for both periods we can observe that there are small differences between winter and summer. In the case of period 1 the maximum difference between winter and summer is 0.9 K and it is observed in the lower part while for period 2

10    the differences are lower than 0.7 K. Much larger differences are found for the standard deviation between the two seasons for period 1 (dashed lines). While the standard deviations ranges between 0.9 K and 1.8 K in summer, the values ranged between 2 K and 4.5 K in winter, reaching the maximum standard deviation at 28.5 km. Although during period 2 the standard deviations in winter were also larger than in summer, the differences were not so remarkable (smaller than 0.5 K). These results show that





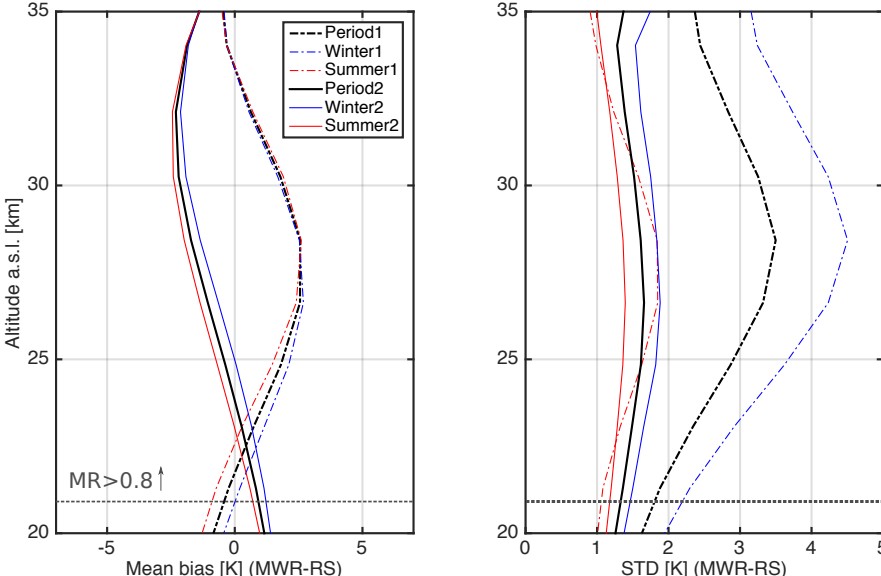

**Figure 8.** Mean temperature and standard deviations between TEMPERA and RS. A total of 1489 profiles have been compared (Period 1: 809 prof., dashed lines; Period 2: 680 prof., solid lines). The mean and the standard deviations for each periods are represented by black lines. The winter season is indicated with blue lines while the summer is indicated by red lines (Winter1: 421 prof.; Summer1: 388 prof.; Winter2: 289 prof. Summer2: 391).

there was a larger variability in the temperature deviations between TEMPERA and RS during the winters of period 1. It is something that could be expected from the temperature evolution showed in Fig. 7 that showed larger discrepancies specially during winter 2015.

## 4.2 Comparison with Aura/MLS

The stratospheric temperature profiles from TEMPERA have also been compared with the ones obtained from the MLS instrument on board of Aura satellite. As it was indicated in section 2 the temperature profiles used for MLS correspond to the version 4 retrievals. In order to select the temperature profiles from MLS to be used in the comparison we chose those ones that where collocated with the measurement site, and for our criterion it was the MLS measurements inside of the range of $\pm 1°$ ($\pm 110$ km) in latitude and $\pm 5°$ ($\pm 460$ km) in longitude. The data were also restricted to cases with near time-coincident

between TEMPERA and MLS. A total of 367 profiles were obtained under these criteria and for all weather conditions excluding rainy cases. The temperature profiles of MLS were interpolated to the pressure grid of TEMPERA and these profiles were convolved using the averaging kernels of TEMPERA as it was described in section 3.

Figure 9 shows the evolution of the stratospheric temperatures and the deviations between TEMPERA and MLS at 3 different altitude levels. Similar patterns to the ones observed in Fig. 7 are found in this plot (although with less data), observing an

annual cycle with higher temperature in summer than in winter and with a larger variability during wintertime. We can observe





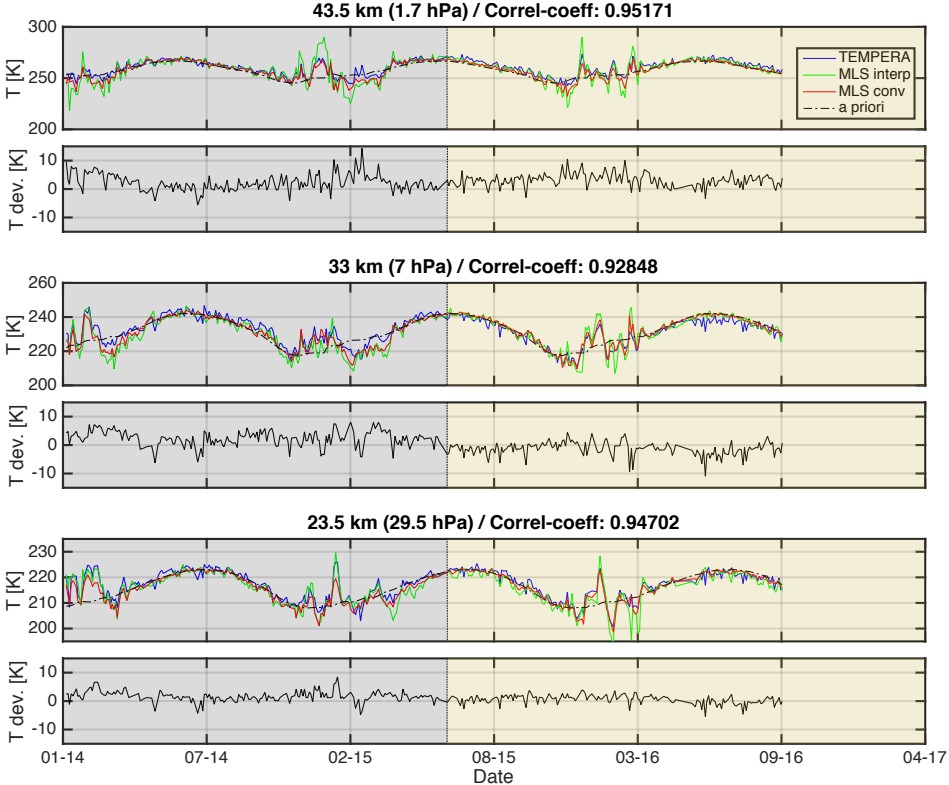

**Figure 9.** Stratospheric temperature evolution and temperature deviations at different altitudes for TEMPERA and MLS. Different background colors are used to distinguish between period 1 and 2 (gray and light brown, respectively).

from these plots a very good agreement between both instruments despite the very different type of observations that we are comparing (ground-based against satellite measurements). This good agreement is also observed when strong variations of temperature are exhibited in a short interval time as it can be seen in winter of 2016 and it is confirmed by the high correlation coefficient (larger than 0.92) found at the different altitudes. The temperature deviations (TEMPERA-MLS) observed are in

5 general small, although we can observe some larger discrepancies for some measurements (reaching deviations of 10 K) mainly during wintertime. Differences between TEMPERA and MLS retrievals can arise from several factors, including differences due to spatio-temporal inhomogeneities due to synoptic variability which can be more important during winter, differences in vertical resolution or interpolation techniques or measurements errors of both instruments.

The mean and the standard deviations between TEMPERA and MLS for all the measurements of the two periods described

10 in the previous section and also for the different seasons have been plotted in Figure 10. From this comparison, a clear change in the mean bias is again observed between both periods in the lower part of the stratosphere (from 20 to 37 km). In that range, the mean bias in the period 1 was $2.5 \pm 1.3$ K reaching a maximum deviation of 4.1 K at 28.5 km while for period 2 the mean bias was $-0.4 \pm 0.9$ K with a maximum negative deviation of $-1.4$ K at 30 km. In the upper part (between 38 and 50 km) the





differences in the biases were not so significant with a mean value of $1.7 \pm 0.5$ K for period 1 and $2.3 \pm 0.7$ K for period 2. The standard deviation shows again higher values for period 1 than for period 2, although the differences were smaller than in the comparison with RS. The mean standard deviations in the range between 20 and 50 km were $2.4\pm0.6$ K for period 1 and $2.0\pm0.4$ K for period 2 .

This comparison also shows a seasonal behaviour for the mean and the standard deviation of the temperature differences between TEMPERA and MLS for both periods. For period 1 there was a positive bias for both seasons almost in the whole column with larger values in winter than in summer. The mean bias in the lower part (20-35 km) was $3.3\pm1.2$ K in winter and $1.9\pm1.4$ K in summer. The discrepancies were even larger in the upper part (35-50 km) showing much lower bias in summer ($0.4\pm0.4$ K ) than in winter ($2.8\pm0.7$ K ). During period 2 the differences between the biases in winter and summer were quite
constant in altitude and they were always lower than 1.6 K. The standard deviations of the temperature differences showed higher values in winter than in summer for both periods. For period 1 the mean standard deviation for the whole range (20-50 km) was $2.5\pm0.5$ K in winter reaching a maximum value (3.1 K) at 28.5 km while for period 2 the mean standard deviation was $2.1\pm0.5$ K with a maximum value of 2.6 K at 32 km. The standard deviations in summer for both periods were very similar with mean values for the whole altitude range (20-50 km) of $1.8\pm0.6$ K in period 1 and $1.7\pm0.5$ K in period 2. These
results show again the lower temperature discrepancies observed between TEMPERA and MLS satellite during summertime. The biases found in this comparison are similar to the ones reported by Schwartz et al. (2008) for a comparison between MLS version 2.2 retrievals and different analyses and observations (GEOS-5, ECMWF, radiosondes, AIRS/AMSU, etc), where the biases ranged between -2.5 K and +1.

The MLS measurements have also been compared with the ones from RS in the range where they were comparable (lower
stratosphere). Only collocated MLS profiles (with the same criteria explained above) and measured in an temporal interval lower than 4 hours to the RS launch were selected for the comparison. A total of 323 pair of profiles fulfilled these criteria and were used for this statistics. The RS profiles were interpolated to the pressure grid of MLS in order to perform the direct comparison of their profiles. Figure 11 shows the mean and the standard deviation for this comparison. We can observe that the mean bias ranges between -1.7 K at 19 km and +1.4 at 19.6 km. The standard deviation of the temperature differences
between MLS and RS was quite constant in altitude with a mean value of $1.7\pm0.2$ K and a maximum standard deviation of 2.2 K reached at 31 km. We can note that the bias and the standard deviation observed between MLS and RS is very similar to the values observed in the comparison between TEMPERA and RS in period 2 (biases ranging between -2.3 K and 0.9 K and the standard deviations between 1.3 K and 1.7 K). The slight underestimation of the temperature in most of the altitudes found for MLS versus RS in this study agrees with the results obtained by Schwartz et al. (2008) between MLS and different sources.

### 4.3   Comparison with lidar measurements

TEMPERA radiometer has also been compared with an active remote sensing instrument, a Rayleigh lidar. This lidar is operated at Hohenpeißenberg station (Germany) which is located around 400 km Northwestward of Payerne. Despite the distance between both instruments, we wanted to evaluate the agreement in the stratospheric temperature between these very different techniques. A total of 192 profiles have been compared for all weather conditions (excepting rainy cases) for the period from





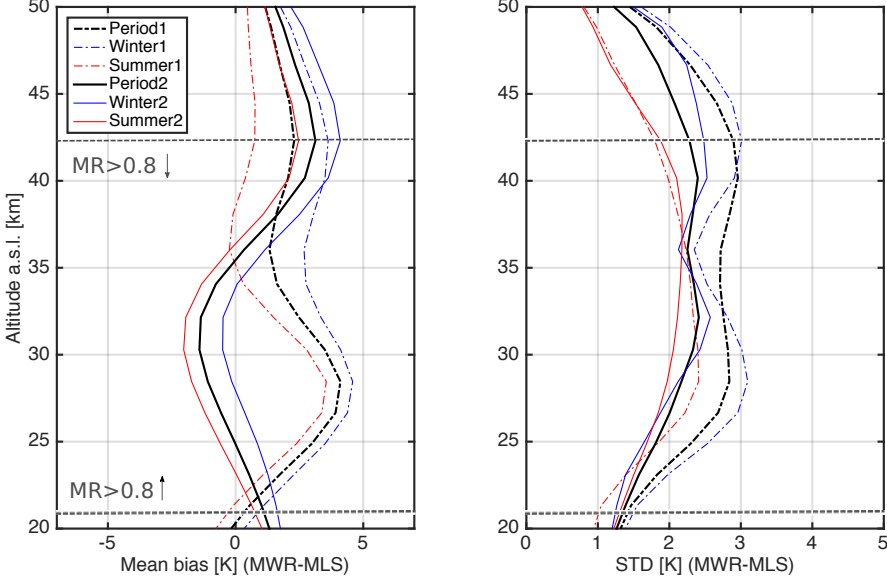

**Figure 10.** Mean temperature and standard deviations between TEMPERA and MLS. A total of 358 profiles have been compared (Period 1: 192 prof., dash lines; Period 2: 166 prof., solid lines). The mean and the standard deviations for each period are represented by black lines. The winter season is indicated with blue lines while the summer is indicated by red lines (Winter1: 103 prof.; Summer1: 89 prof.; Winter2: 67 prof. Summer2: 99).

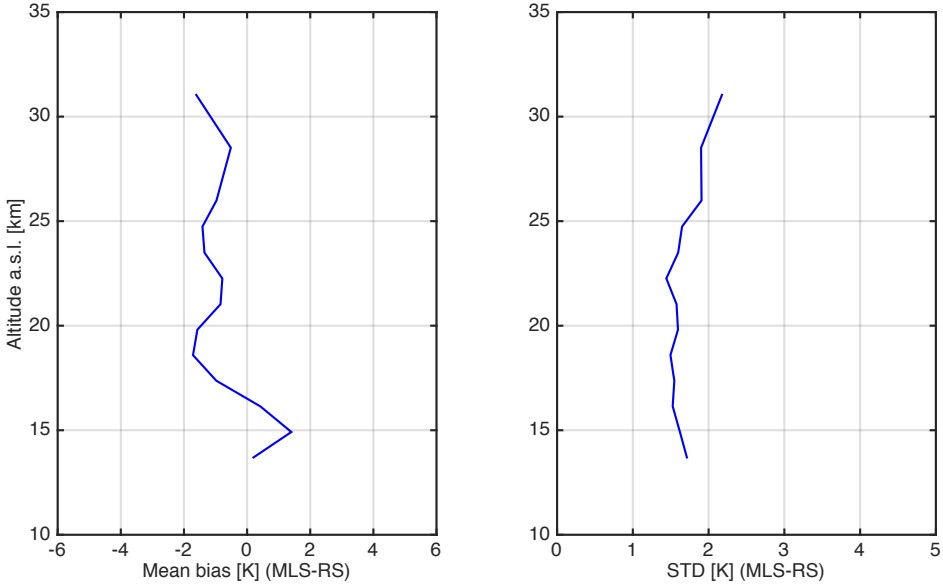

**Figure 11.** Mean and standard temperature deviation between MLS and RS.





January 2014 to July 2016. As in the previous comparisons the lidar profiles were also interpolated to the pressure grid of TEMPERA radiometer and then these profiles were convolved using the averaging kernel of TEMPERA. Since the Rayleigh lidar only provides temperature information above approximately 28 km (below the measurements would be affected by strato-spheric aerosol), the gap below this altitude was filled with coincident measurements from TEMPERA in order not to modify

the averaging kernel used by TEMPERA for the convolution.

Figure 12 shows the stratospheric temperature evolution from TEMPERA and the lidar at three different altitude levels. For the lowest altitudes shown here (29.5 km, asl), the temperature from RS has also been plotted since at this altitude there were measurements from the three instruments. We can observe from this figure that there is a good agreement between TEMPERA and the lidar in the upper stratosphere, with correlation coefficients larger than 0.94 for the two highest altitudes.

This coefficient is lower (0.9) for the lowest altitude (29.5 km, asl). The agreement between the lidar and the RS in this lowest altitude is better than for TEMPERA with a correlation coefficient of 0.96. The evolution of the temperature deviations between TEMPERA and lidar at the three altitudes shows small discrepancies for both techniques along the years, with the values in most measurements below 5 K. The biggest differences were found at the lowest altitude (29.5 km, asl), where a clear change of bias was observed after summer 2015.

Figure 13 shows the mean and the standard deviation for all the measurements in periods 1 and 2 in addition to seasonal profiles. Mean bias profiles show again a clear change in the tendency of the biases of both periods, being more evident in the lower stratosphere (below 35 km). In this lowest altitude range the mean biases were 2.7±1.3 K for period 1 and -1.2±0.4 K for period 2. Above 35 km the differences between the biases were smaller with a largest bias for period 2 (2.3±0.9 K versus 1.3±0.4 K in period 1). Similar behaviour to the other comparisons has been observed for the standard deviation with largest

values during period 1 than during period 2. The mean values for the whole altitude range were 2.9±0.3 K for period 1 and 2.5±0.2 K for period 2. A seasonal behaviour is observed in the bias and standard deviation for both periods. The seasonal biases showed a vertical oscillation with different tendencies for both periods in the lower and upper part of the stratosphere. For the lower part (28-35 km) the mean biases for period 1 (period 2) were 3.2±1.1 K (-0.7±0.4 K) in winter and 1.9±1.5 K (-2.1±0.3 K) in summer. In the upper part (35-50 km), a general positive bias was observed between TEMPERA and the lidar

where the mean biases for period 1 (period 2) were 2.2±0.6 K (2.9±1.1 K) in winter and -0.3±0.3 K (1.1±0.9 K) in summer. The standard deviations showed larger values in winter for both periods than in summer. The highest standard deviations were again observed in winter of period 1. The mean standard deviations in the whole column were for period 1 (period 2) 3.1±0.4 K (2.6±0.3 K) in winter and 2.0±0.3 K (1.7±0.4 K) in summer.

## 4.4   Comparison with SD-WACCM

A first validation of the stratospheric temperature from SD-WACCM (Whole Atmosphere Community Climate Model with Specified Dynamics) has also been carried out in this study. SD-WACCM is the whole atmosphere component of CESM (Community Earth System Model) (Kunz et al., 2011; Lamarque et al., 2012). CESM is a coupled climate model which means that it consists of separate models for different parts of the climate system which interact via the coupler module. There are





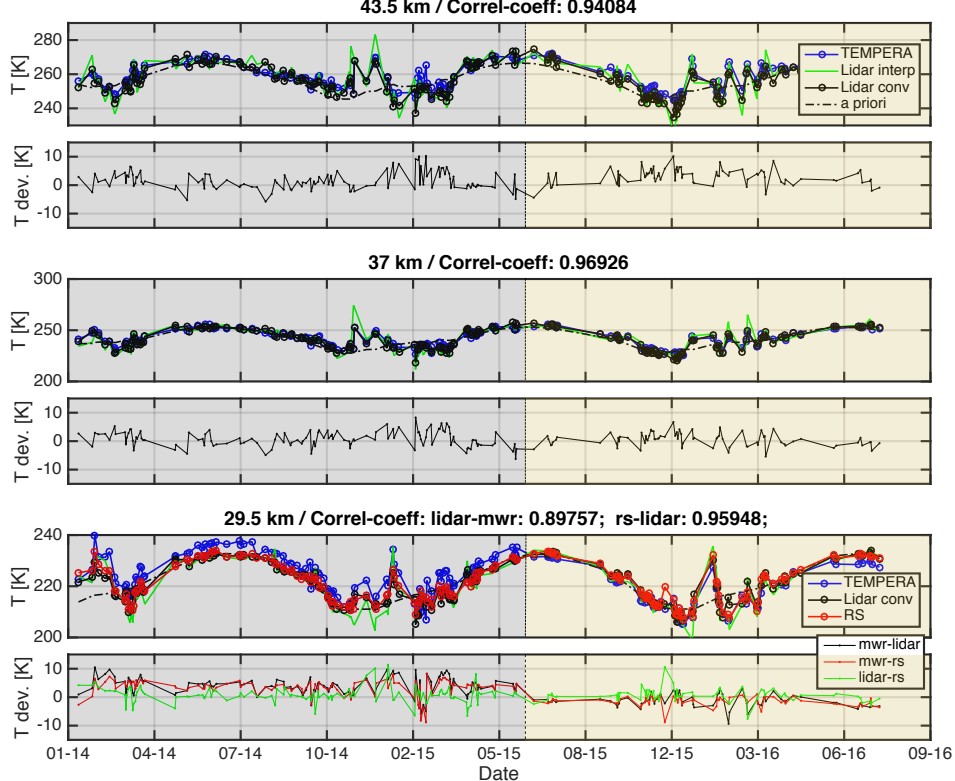

**Figure 12.** Stratospheric temperature evolution from TEMPERA, lidar and RS. Different background colors are used to distinguish between period 1 and 2 (gray and light brown, respectively).

models for ocean, atmosphere, land, sea ice, land ice and rivers. CESM allows to combine the above models to a component set for the simulation.

The Specified Dynamics (SD) used in these simulations means that the model is nudged by meteorological analysis fields by 10% at every internal time-step up to an altitude of 50 km. This means 90% of the model and 10% of the nudging data are taken. The fields that are nudged are temperature, horizontal winds, surface wind stress, surface pressure and heat fluxes from the surface. The nudging data is from the Goddard Earth Observing System version 5.0.1 (GEOS-5) Data Assimilation and is provided every 6 hours, in between the data are interpolated.

The altitude range for SD-WACCM is from ground to 140 km (asl). The altitude resolution ranges from 0.5 to 4 km (with lower resolution at higher levels) and with a total of 88 layers in the whole atmosphere. The grid resolution is 1.9° latitude by 2.5° longitude.

The stratospheric temperatures from SD-WACCM have been compared with the almost continuous stratospheric temperature profiles measured by TEMPERA radiometer for the period from January 2014 to April 2016. A total of 6868 profiles were selected under all weather conditions except rainy conditions. Figure 14 shows the stratospheric temperature evolution along





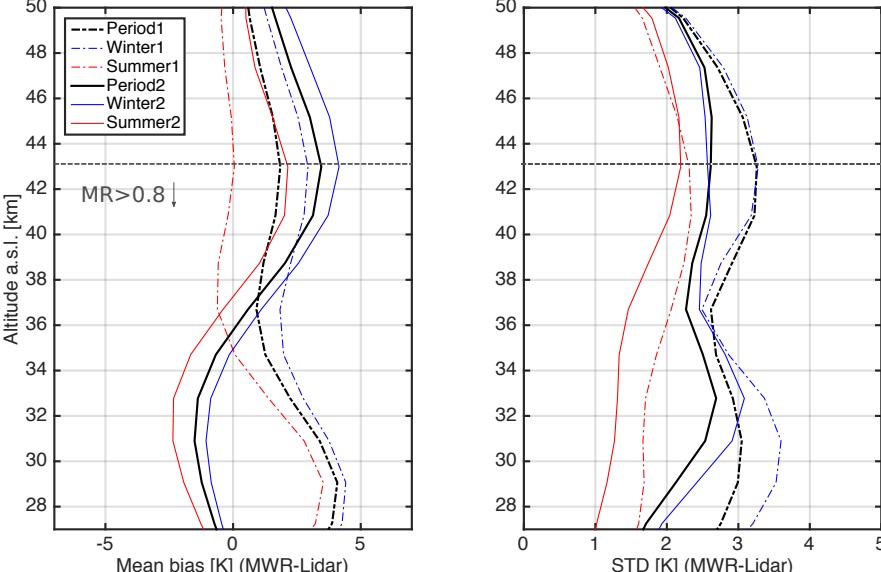

**Figure 13.** Mean temperature deviation between TEMPERA and lidar. A total of 192 profiles have been compared (Period 1: 117 prof., dash lines; Period 2: 75 prof., solid lines). The mean and the standard deviations for each period are represented by black lines. The winter season is indicated with blue lines while the summer is indicated by red lines (Winter1: 73 prof.; Summer1: 44 prof.; Winter2: 49 prof. Summer2: 26).

this period for TEMPERA and WACCM. A good agreement is observed in general between both temperature sets. We can observe as the temperature from the model follows the same pattern as for TEMPERA, with the same annual cycle and detecting the same structures in time and also in altitude. It is worth to point out the good agreement observed during winters, where strong increases of temperatures are produced for short periods and can be observed in both data sets. The differences

between TEMPERA and WACCM are more evident above 50 km (asl), but above this altitude the measurement response for TEMPERA is low (lower than 0.6) since the weight of the measurements is small and should not be considered in the comparison.

The temperature profiles from SD-WACCM have been interpolated and convolved as it was explained in section 3 in order to compare with the ones from TEMPERA. Figure 15 shows the evolution of the temperature at three altitude levels and

the differences between both (TEMPERA-WACCM). The good agreement observed from these plots is proven by the low temperature deviation values along the time (lower than 5 K in most of the time) and the large correlation coefficient (larger than 0.92). Despite this good agreement, we also find some periods with larger discrepancies between the measurements and the model, specially during winter time and being more evident in winter 2015. It is worth to point out the very robust statistics that we are showing in this section, with almost 7000 pairs of temperature profiles compared.

We have also calculated the bias and the standard deviation for this comparison between TEMPERA radiometer and WACCM model (Fig. 16). It is again very obvious from this statistics the strong change in the biases between periods 1





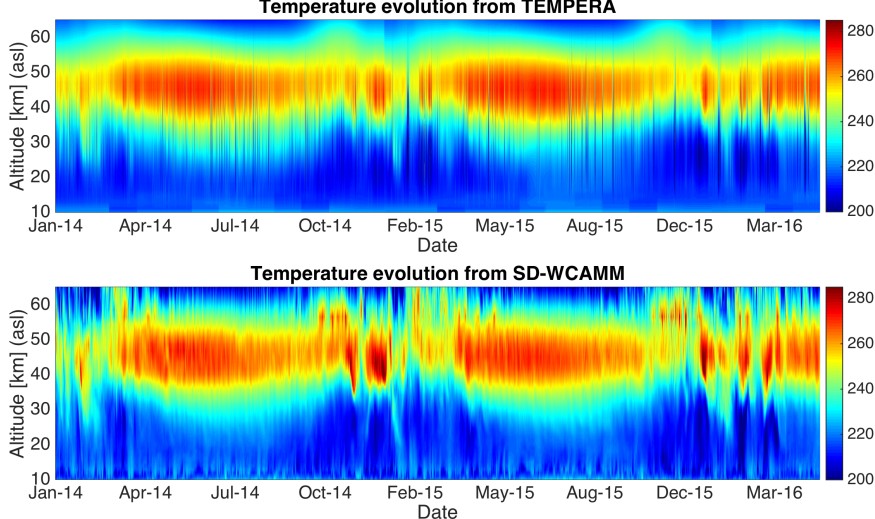

**Figure 14.** Stratospheric temperature from TEMPERA radiometer (upper panel) and WACCM model (lower panel).

and 2 with a very different tendency in the lower stratosphere. The mean biases for the lower part (20-35 km) were $1.4\pm1.1$ K for period 1 and $-1.0\pm1.3$ K for period 2. While the mean biases for the upper stratosphere (35-50 km) were $1.0\pm0.7$ K for period 1 and $1.7\pm1.1$ K for period 2. It is worth to remark the almost negligible seasonal behaviour observed in the biases for both periods.

From the standard deviation figure (Fig. 16, right) we can observe that much larger values are obtained for period 1 with a mean value in the whole column of $2.9\pm0.6$ K and a maximum standard deviation of 3.8 K at 29 km. This large standard deviations observed during period 1 is strongly influenced by the large values observed during winter time (blue dashed line) that reached a maximum standard deviation of 4.7 K at 29 km. The rest of the standard deviation profiles show very similar values between them increasing slightly in the lower part (up to 30 km) and keeping close to constant values above this altitude.

The smallest values are found in summer with a mean bias in the whole column of $1.8\pm0.4$ K for period 1 and $1.5\pm0.3$ K for period 2.

### 4.5 All measurements and model versus TEMPERA

In order to summarize the intercomparison carried out between TEMPERA and the different measurement techniques and model we have plotted together the biases and the standard deviations for all the comparisons (Fig. 17). Since we are interested

in evaluating the accuracy and precision of TEMPERA radiometer against other measurements in this study we have only displayed in Fig. 17 the biases and the standard deviations obtained for the summer season which is less affected by atmospheric variability than wintertime.

The mean bias plot (Fig. 17, left) shows a clear change of biases between TEMPERA and all the other measurements for the first (dashed lines) and the second (solid lines) period (before and after the repair of the FFT spectrometer's attenuator).



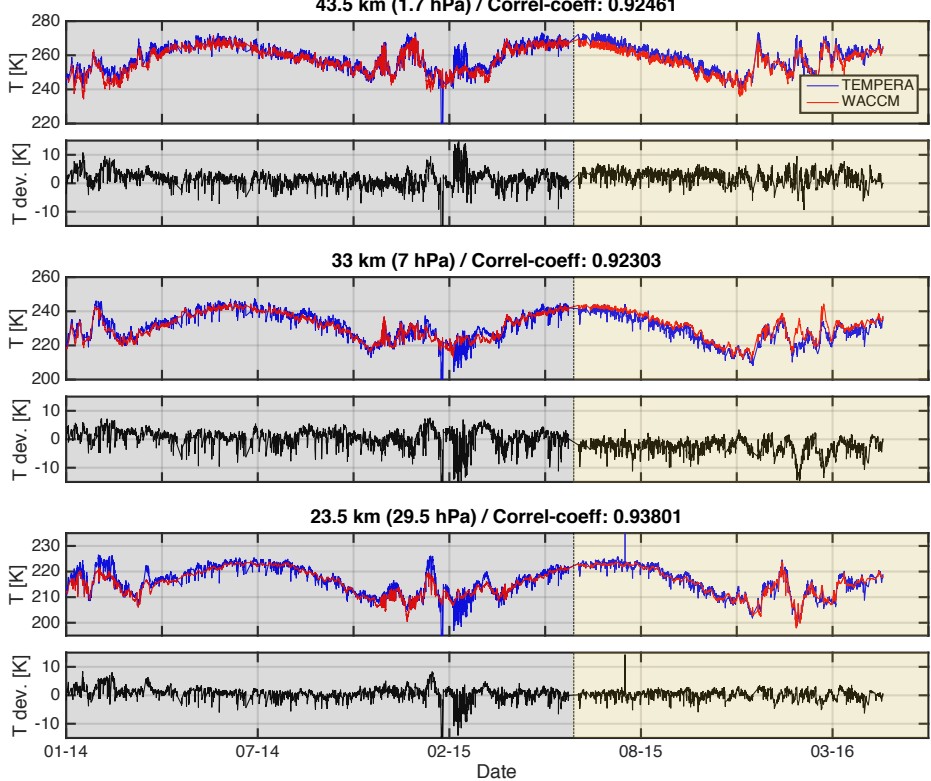

**Figure 15.** Stratospheric temperature evolution from TEMPERA and WACCM. Different background colors are used to distinguish between period 1 and 2 (gray and light brown, respectively).

We can observe that there is a persistent vertical oscillation for all the profiles in both periods causing a different behaviour of the biases in the lower and upper stratosphere. This oscillation has an amplitude of around 2 K and a periodicity of roughly 20 km. Similar behaviour was observed for MLS measurements when they were compared with different sources (Schwartz et al., 2008). The change of tendency in the bias between both periods is more evident in the lower stratosphere (below 35 km) where

5  we can observe that almost for all the altitude levels the biases changed from positive to negative values in all the comparisons. Another remarkable point is the consistency between the different biases in each period, showing small differences between them (below 1K) for most of the altitudes, specially for period 2. For period 1, the maximum deviations were found at 28.5 km, with a maximum value of 3.6 K for the comparison with MLS satellite. Below this altitude, an almost identical bias between the comparison with RS and WACCM model is found. In the upper stratosphere the biases were between -0.6 K and 1.5 K showing

10  the lowest bias with the lidar. For period 2 the values of the different biases ranged between -2.4 K (at 32 km) and a maximum positive bias of 2.9 K (at 43 km) was found for the comparison with WACCM. As we already mention the differences between the different comparisons for period 2 were smaller than for period 1 showing the consistency between the RS, MLS, lidar measurements and also WACCM simulations.





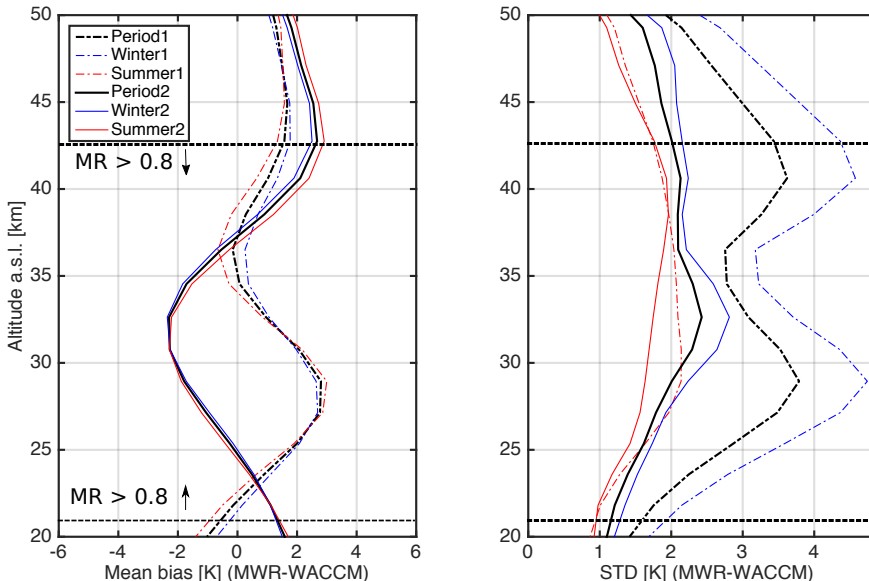

**Figure 16.** Mean temperature deviation between TEMPERA and WACCM model. A total of 6868 profiles have been compared (Period 1: 4339 prof., dash lines; Period 2: 2529 prof., solid lines). The mean and the standard deviations for each period are represented by black lines. The winter season is indicated with blue lines while the summer is indicated by red lines (Winter1: 2361 prof.; Summer1: 1978 prof.; Winter2: 1473 prof. Summer2: 1056).

Fig. 17(right) shows the standard deviations of the differences between TEMPERA and the different measurements and model. In general we can observe that there was a reduction in the standard deviations for all the comparisons in period 2, indicating that the precision of TEMPERA improved after the attenuator was repaired. Next, we will focus our discussion on period 2, when we consider that TEMPERA was operated in its optimal status. For this period, we can observe that standard
deviations were always lower than 2.2 K and this maximum values was reached at 45 km for the comparison with lidar. The lowest standard deviation in the lower stratosphere was found for the comparison with RS and the mean value of the standard deviation in this range was 1.3±0.1 K. The highest standard deviations in the lower stratosphere was found for the comparison with MLS (1.3±0.1 K). These results evidence a better precision of TEMPERA radiometer when it is compared with the in situ reference technique of RS in the lower stratosphere. This result makes sense since RS is the technique with the lowest errors
(0.2 K for temperature) and the comparison between TEMPERA and RS is the one that should present lower atmospheric variability between both measurements since RS are launched in the same location that TEMPERA is operated.

In the middle stratosphere (between 30 and 40 km) the lowest standard deviations were found for the comparison with lidar with a mean value of 1.4±0.2 K. However, above this altitude the deviations with lidar are the largest (1.4±0.2 K). In this upper part the lowest standard deviations were found for the comparison with MLS and WACCM. A common pattern that is observed
in all the comparisons is that the standard deviations decrease slightly with altitude in the last kilometres of the stratosphere. This behaviour is due to a greater weight of the a priori temperature profile used in the TEMPERA retrievals and also in the





**Figure 17.** Mean and standard temperature deviations between TEMPERA radiometer and the measurements from the different instruments and WACCM.

convolved profiles in these altitudes since the measurement response presents lower values for high altitudes (around 0.6 at 48 km).

Table 1 presents the different biases and standard deviations obtained in the lower and upper stratosphere for all the comparisons during summertime in period 2. These values could be considered as the most representative way of characterizing the
accuracy and precision of TEMPERA radiometer. Since they correspond to the period when TEMPERA was running with the repaired attenuator (period 2) and also when the measurements where less affected by atmospheric variability (summertime).

We would like to end by highlighting the consistency found between the standard deviations of the different comparisons and the observation errors of TEMPERA retrievals. As we already mentioned in section 3.1 the OEM also estimates the observation, smoothing and total errors of the TEMPERA inversions (Fig. 4e). The standard deviations found in the different comparisons
are somehow related to the observation error of TEMPERA but also to the errors associated with the other measurements and the atmospheric mismatches. If we assume that the random errors in TEMPERA (D1), in the other instruments (D2) and in the atmospheric mismatching (D3) are independent, then the observed standard deviation (DT) should be given by $DT^2 = DT1^2 + DT2^2 + DT3^2$. For example, if we consider the observation error of TEMPERA provided by OEM (0.8 K), the errors for lidar (0.7 K) and the mean observed standard deviation for the comparison between TEMPERA and lidar (1.1 K) we would
conclude that the errors associated with atmospheric mismatches should be 0.3 K which is a realistic value and in this way shows the consistency between the observed standard deviations and the observation errors of the different measurements.



**Table 1.** Range of bias and standard deviations between RS, MLS, lidar and WACCM with TEMPERA radiometer.

|  |  | MWR-RS | MWR-MLS | MWR-lidar | MWR-WACCM |
|---|---|---|---|---|---|
| lower strat. | BIAS | -1.3±1.1 | -1.0±1.0 | -1.1±1.3 | -1.0±1.3 |
| (20-35 km) | STD | 1.3±0.1 | 1.8±0.3 | 1.1±0.2 | 1.5±0.3 |
| upper strato. | BIAS |  | 1.5±0.9 | 1.1±0.9 | 1.9±1.1 |
| (35-50 km) | STD |  | 1.7±0.5 | 1.9±0.3 | 1.6±0.3 |

## 5  Conclusions

Almost three years of measurements of stratospheric temperature profiles from a relatively new ground-based microwave radiometer (TEMPERA) have been intercompared with the ones from different measurement techniques as they are RS, MLS satellite and Rayleigh lidar and also from SD-WACCM model. TEMPERA measurements were carried out at the aerological station of MeteoSwiss at Payerne from January 2014 to September 2016. Ground-based microwave measurements present as main advantages that they can provide unattended continuous measurements of temperature profiles in almost all weather conditions with a reasonable good spatial and temporal resolution. The stratospheric temperature profiles (from 20 to 50 km) were obtained from TEMPERA measurements using OEM by means of the radiative transfer model ARTS/QPack. All the profiles from the other techniques (RS, MLS and lidar) and from WACCM model were interpolated to the TEMPERA pressure grid and then convolved using the averaging kernel of this radiometer in order to compared with the ones from TEMPERA.

The temperature evolutions at different altitudes of TEMPERA and the different measurements and model showed in general a very good agreement with a high correlation (always larger than 0.9) between the compared datasets. These stratospheric temperature evolutions showed a larger variability during wintertime and also evidenced larger discrepancies between TEMPERA and the other datasets during that periods. A small step in the temperature deviations was observed in July of 2015 for the different comparisons, and it was related with the repair of an attenuator in the FFT spectrometer of TEMPERA. This repair caused a small modification in the measured brightness temperature from TEMPERA and therefore in the retrieved temperature profile from this radiometer. For this reason, and in order to take into account this instrumental modification and characterize possible changes in the accuracy and precision of TEMPERA radiometer the statistical analysis was carried out over two different measurement periods (before and after the modification). In addition a seasonal distinction (winter and summer) was considered in the statistics to take into account the larger atmospheric variability that could be observed during wintertime and which could produce larger deviations between the instruments due to the atmospheric conditions.

The accuracy and the precision of TEMPERA radiometer have been evaluated by means of the bias and the standard deviation between TEMPERA and other measurements and model outputs (RS, MLS, lidar and WACCM). The stratospheric temperature comparison between TEMPERA and the other datasets showed a clear change in the biases between periods 1 and 2 (before and after the repair of the attenuator) in all the statistics. For the lower stratosphere (20-35 km) the biases changed from positive values in period 1 to negative values in period 2. The smallest mean deviations were observed in the compar-





ison with RS with values always lower than $\pm$ 2.5 K. The largest biases were observed for the comparisons with MLS and the Rayleigh lidar reaching maximum deviations of around +4.5 K at some altitudes in period 2. In general the biases were smaller and with negative sign for all the comparisons during period 2, indicating a slight underestimation of the temperature by TEMPERA radiometer in that period.

5     In the upper part of the stratosphere (above 35 km) the differences between both periods were not so evident, and general positive biases were observed in both periods for all the comparisons. The deviations in this upper part were always lower than 4.5 K. We would like to point out the weak seasonal behaviour observed for the biases in the comparisons with RS and WACCM while it was more pronounced for the comparison with MLS and lidar, specially in period 1.

    The standard deviations obtained from the different statistics showed again very different results between both periods. 10   Larger values were observed for all the comparisons in period 1 indicating that the precision of TEMPERA radiometer improved after the repair of the spectrometer's attenuator. The standard deviations were especially high in wintertime of period 1 reaching maximum values of around 4.5 K for the comparison with RS (at 28 km) and MLS (28 km and 41 km). In period 2 the standard deviations during winter were also larger than in summer but with smaller differences (except for the lidar in the lower part). These results confirmed the larger atmospheric variability that can be found during wintertime and which produce 15   a lower agreement in the temperature measurements between the different instruments, especially when the horizontal distance between them is large.

    Finally, the accuracy and the precision of TEMPERA radiometer have been characterized by means of the bias and the standard deviation of this radiometer versus the different measurements and model obtained during period 2 (instrument in optimal conditions) and in summer (less affected by atmospheric variability). This statistics in the lower stratosphere (below 20   35 km) showed mean biases ranging between 1.0 and 1.3 K (max. for RS and min. for MLS) and mean standard deviations that ranged between 1.1 and 1.8 K (max. for MLS and min. for lidar). While in the upper stratosphere (above 35 km) the mean biases ranged between 1.1 and 1.9 K (max. for WACCM and min. for lidar) and the mean standard deviations ranged between 1.6 and 1.9 K (min. for WACCM and max. for lidar). The standard deviations observed in the different comparisons were consistent with the observation errors that are expected from the different instruments indicating that it is a good measure of 25   the instrumental errors.

    From all these intercomparisons we can conclude that TEMPERA radiometer has shown a very good performance to determine the temperature in the stratosphere. Continuous TEMPERA measurements will allow in the future to carry out temperature trend analysis which are an important component of the global change. These trends can provide evidence of the roles of natural and anthropogenic climate change mechanisms. Stratospheric temperature changes are also crucial for understanding 30   stratospheric ozone variability and trends, including predicting future changes. In addition, measurements with a high temporal resolution in a fixed location will also allow to characterize the local thermodynamics which can be specially interesting during wintertime.





*Acknowledgements.* We thank MeteoSwiss and in particular Dominique Ruffieux, Ludovic Renaud, Philippe Overney and Jean-Marc Aellen for hosting our instrument and for the support on-site. This work has been funded by the Swiss National Science Foundation under grant 200020-160048 and MeteoSwiss in the framework of the GAW project "Fundamental GAW Parameters by Microwave Radiometry".



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
