# Peer review of "Intercomparison of stratospheric temperature profiles from a ground-based microwave radiometer with other techniques"

_Atmospheric Chemistry and Physics, 2017_

## Referee Comment (RC1) · Anonymous Referee #1 · 27 Jun 2017

review uploaded as a supplement

Please also note the supplement to this comment:
https://www.atmos-chem-phys-discuss.net/acp-2017-346/acp-2017-346-RC1-supplement.pdf

———————————————

[Figure]

This manuscript shows some very nice measurements of temperatures in the stratosphere. The measurements compare well with various other sources. While I have several suggestions, I have no major recommendations for changes. My most serious complaint is that, while many of the authors of this manuscript speak excellent English, much of the text is very poorly written. I certainly do not think that it should be the reviewer's role to assist in this task, especially when several of the co-authors are completely capable of doing so.

Below are a series of specific suggested changes:

Page 5 - "Radiosondes reach an altitude of 35km". This, and later statements, gives the impression that all radiosondes from Payerne reach precisely this altitude, but I am skeptical that this is the case.

Figure 3 – Given that, according for Figure 4d, the measurement response falls to well below 50% at ~17km, and that, as far as I have been able to determine, other TEMPERA studies show sensitivity only up to at best ~7km, suggesting that TEMPERA measures into the upper troposphere is very deceptive. It does certainly not, as the text suggests "cover the full troposphere and stratosphere".

Figure 4b – This Figure is a simplification of a very similar Figure 14 in Stähli et al. (2013). In that figure it is stated that "In the center of the lines we use all channels and on the wings of the line we use a binning of 3 channels for data reduction". I am almost certain that this is why the middle channels are noisier, and that this has nothing to do with the Zeeman effect, as is stated in the paper.

Figure 4c – What is meant by "observation error"? Given that there are systematic changes >2K in the dataset, I assume that this must be some kind of random error estimate. If this is the case please label it as such. How is it calculated?

Figure 6 – Given the very large discrepancy between the MLS and other measurements on 4 February above 35km, this clearly warrants some discussion. It is very troubling that neither the LIDAR nor the MWR show the decrease in temperature above 35km. Do nearby (in time and space) MLS profiles show the same structure? Do the authors think that this is a bad MLS profile?

Page 9 – "The measurements presented in the plot show the importance of continuous observations for a fixed location, since the variability of atmospheric parameters such as temperature evinces the necessity of measurements with good temporal resolution." This suggests that there are temperature variations every few hours (a conclusion that can certainly not be reached by looking at Figure 5). If this is the case, please show such. If not, then certainly daily satellite measurements must be adequate, and this statement should be removed.

Page 11 – It is stated that, above 35km the RS profiles were extrapolated using TEMPERA profiles. But the top altitude of RS profiles varies, so exactly what does this mean? Were only RS profiles which reached 35km included. If there was data above this was it included (instead of using the TEMPERA data)?

Figures 7, 9, and 12 – The most piece of information on these is the comparison between TEMPERA and the convolved retrievals from the other instruments. Since this is the case, it would be best to plot the

**Fig. 1.**

**Supplement:**

This manuscript shows some very nice measurements of temperatures in the stratosphere. The measurements compare well with various other sources. While I have several suggestions, I have no major recommendations for changes. My most serious complaint is that, while many of the authors of this manuscript speak excellent English, much of the text is very poorly written. I certainly do not think that it should be the reviewer's role to assist in this task, especially when several of the co-authors are completely capable of doing so.

Below are a series of specific suggested changes:

Page 5 - "Radiosondes reach an altitude of 35km". This, and later statements, gives the impression that all radiosondes from Payerne reach precisely this altitude, but I am skeptical that this is the case.

Figure 3 – Given that, according for Figure 4d, the measurement response falls to well below 50% at ~17km, and that, as far as I have been able to determine, other TEMPERA studies show sensitivity only up to at best ~7km, suggesting that TEMPERA measures into the upper troposphere is very deceptive. It does certainly not, as the text suggests "cover the full troposphere and stratosphere".

Figure 4b – This Figure is a simplification of a very similar Figure 14 in Stähli et al. (2013). In that figure it is stated that "In the center of the lines we use all channels and on the wings of the line we use a binning of 3 channels for data reduction". I am almost certain that this is why the middle channels are noisier, and that this has nothing to do with the Zeeman effect, as is stated in the paper.

Figure 4c – What is meant by "observation error"? Given that there are systematic changes >2K in the dataset, I assume that this must be some kind of random error estimate. If this is the case please label it as such. How is it calculated?

Figure 6 – Given the very large discrepancy between the MLS and other measurements on 4 February above 35km, this clearly warrants some discussion. It is very troubling that neither the LIDAR nor the MWR show the decrease in temperature above 35km. Do nearby (in time and space) MLS profiles show the same structure? Do the authors think that this is a bad MLS profile?

Page 9 – "The measurements presented in the plot show the importance of continuous observations for a fixed location, since the variability of atmospheric parameters such as temperature evinces the necessity of measurements with good temporal resolution." This suggests that there are temperature variations every few hours (a conclusion that can certainly not be reached by looking at Figure 5). If this is the case, please show such. If not, then certainly daily satellite measurements must be adequate, and this statement should be removed.

Page 11 – It is stated that, above 35km the RS profiles were extrapolated using TEMPERA profiles. But the top altitude of RS profiles varies, so exactly what does this mean? Were only RS profiles which reached 35km included. If there was data above this was it included (instead of using the TEMPERA data)?

Figures 7, 9, and 12 – The most piece of information on these is the comparison between TEMPERA and the convolved retrievals from the other instruments. Since this is the case, it would be best to plot the

interpolated line first, and then the convolved and TEMPERA lines on top of this. As the plots are currently shown it is sometimes difficult to tell whether the TEMPERA line coincides with the convolved or the interpolated measurement. Alternatively, the interpolated measurement could even be dropped from these plots.

Figures 7 and 9 – is the black deviation line TEMPERA vs. RS interp or RS conv? In fact, throughout much of the text it is not clear whether convolved or unconvolved RS and/or MLS data is being used.

The step in the data shown in Figure 7 very helpful in that it provides a useful measure of the systematic errors in these retrievals. I applaud the investigators for not homogenizing the data between the two periods.

Page 13 – "near time-coincident". Does this mean that the MLS profile was taken during the period of spectral integration for the TEMPERA measurement? If so, please state this. If not, please state what exactly "near time coincident" means. The same applies to the RS comparisons.

Page 15 and Figure 11 – The authors note that: "the bias and the standard deviation observed between MLS and RS is very similar to the values observed in the comparison between TEMPERA and RS in period 2." If do not know, and it seems to be nowhere stated, whether in Figures 8 and 10, the MLS and RS measurements are convolved before comparison with the MWR. If they are (and I think they should be), then the appropriate comparison in Figure 11 would be convolved MLS with convolved RS profiles. This could be added as a dashed line in Figure 11.

Figure 17 – The legend is a bit confusing. Please make the 4 instrument lines solid and thick enough so that one can distinguish lidar and WACCM. Then separately show two styles of lines, one for Period 1 and one for Period 2.

Table 1 – Since Period 1 and Period 2 are presented everywhere else, why is only Period 2 in this table?

---

## Referee Comment (RC2) · Anonymous Referee #2 · 31 Jul 2017

The temperature measurements presented performed by both the Tempera instrument and by radiosondes and MLS really shows the great potential of ground based microwave remote sensing. The Bern group has achieved a unique know-how of developing state of the art radiometers for atmospheric measurements and the Tempera instrument is no exception. The paper is very interesting and well structured and gives an important contribution to the microwave remote sensing community. I recommend the paper for publication in ACP.

---

## Author Comment (AC1) · 16 Oct 2017

**Response to referee 1**

We are very grateful to referee 1 for the careful reading of our manuscript and for providing constructive comments which helped to improve the manuscript. This document includes all the referee's comments as well as our replies to every one of them.

**General comment from the referee**

This manuscript shows some very nice measurements of temperatures in the stratosphere. The measurements compare well with various other sources. While I have several suggestions, I have no major recommendations for changes. My most serious complaint is that, while many of the authors of this manuscript speak excellent English, much of the text is very poorly written. I certainly do not think that it should be the reviewer's role to assist in this task, especially when several of the co-authors are completely capable of doing so.

**Author's response:**
We appreciate the positive feedback from the reviewer. Regarding the English writing we have made a detailed review of the text and it has been substantially improved in the new revised manuscript.

**Specific comments**

**1. Comments from the referee:**
Page 5 – "Radiosondes reach an altitude of 35km". This, and later statements, gives the impression that all radiosondes from Payerne reach precisely this altitude, but I am skeptical that this is the case.

**Author's response:**
We agree with the referee that in the way that the sentence was written sounds that RS always reach that altitude, which is certainly not true. In fact, the target altitude of RS is 10 hPa (~32 km), but many go a bit higher, as high as 35 km. For this reason, we have rephrased the sentence in the way that can be read below.

**Author's changes in the manuscript:** p. 5, line 9
"The target level of radiosondes is 10 hPa (approx.. 32 km), and hence cover only the lower stratosphere."

**2. Comments from the referee:** Figure 3 – Given that, according for Figure 4d, the measurement response falls to well below 50% at ~17km, and that, as far as I have been able to determine, other TEMPERA studies show sensitivity only up to at best ~7km, suggesting that TEMPERA measures into the upper troposphere is very deceptive. It does certainly not, as the text suggests "cover the full troposphere and stratosphere".

**Author's response:**
We agree with the referee that the measurement response of TEMPERA is not high enough in the full range from ground to the stratopause. For this reason, we have updated Fig. 3 showing that TEMPERA does not cover the upper troposphere and the lower stratosphere. We have also slightly modified the sentence indicated by the referee in order to be more precise.

**Author's changes in the manuscript:** p. 6, line 8
"As we can see TEMPERA is the only instrument that is able to cover almost the full troposphere and stratosphere."

**3. Comments from the referee:** Figure 4b - This Figure is a simplification of a very similar Figure 14 in Stähli et al. (2013). In that figure it is stated that "In the center of the lines we use all channels and on the wings of the line we use a binning of 3 channels for data reduction". I am almost certain that this is why the middle channels are noisier, and that this has nothing to do with the Zeeman effect, as is stated in the paper.

**Author's response:**
We thank the referee for this clarification. We agree with the referee that most of the noise observed for the residuals in the central part of the lines comes from different binning used. The Zeeman effect could be responsible of some differences between the measured spectra and the modelled ones (the model does not include Zeeman effect), but the differences should be smaller than the ones observed in this Figure. We have clarified this point in the manuscript in the way that can be read below.

**Author's changes in the manuscript:** p. 8, lines 12-15
"The larger differences observed in the center of the emission lines (see Fig. 4b) is mainly due to a different binning used in the center of the lines and on the wings of the lines (Stähli et al., 2013). In addition, the Zeeman effect could explain some small differences in the center of the lines since it is not incorporated in the forward model (Navas-Guzmán et al. 2015)."

**4. Comments from the referee:** Figure 4c - What is meant by "observation error"? Given that there are systematic changes >2K in the dataset, I assume that this must be some kind of random error estimate. If this is the case please label it as such. How is it calculated?

**Author's response:**
The observation error is the error of the retrieved profile due to measurement noise, i.e. the random error. It is calculated by propagating the measurement uncertainty (the measurement noise) through the retrieval using Gaussian error propagation.

**Author's changes in the manuscript:** p. 8, line 23
"Finally, the total, observational (random error due to measurement noise) and smoothing errors are also calculated with this method and are shown in Fig. 4e."

**5. Comments from the referee:** Figure 6 - Given the very large discrepancy between the MLS and other measurements on 4 February above 35 km, this clearly warrants some discussion. It is very troubling that neither the LIDAR nor the MWR show the decrease in temperature above 35km. Do nearby (in time and space) MLS profiles show the same structure? Do the authors think that this is a bad MLS profile?

**Author's response:**
We have double-checked that the nearby (in time and space) MLS profiles also show the same structure than the one presented in the Figure 6a. Since this structure is not detected by the other instruments (TEMPERA and lidar) we think that these could be problematic MLS inversions. Anyway, the idea of presenting this figure in the manuscript was just to show the different altitude ranges and spatial resolutions of the instrumentation used in our study. The observed differences between instruments in these three individual cases evidence the importance of our statistical analysis to really characterize the performance of the different instruments.

**6. Comments from the referee:** Page 9 - "The measurements presented in the plot show the importance of continuous observations for a fixed location, since the variability of atmospheric parameters such as temperature evinces the necessity of measurements with good temporal resolution." This suggests that there are temperature variations every few hours (a conclusion that can certainly not be reached by looking at Figure 5). If this is the case, please show such. If not, then certainly daily satellite measurements must be adequate, and this statement should be removed.

**Author's response:**
According to the referee's suggestion we have added a new plot to Figure 5 in order to show the high variability that temperature can have in the stratosphere in the course of few hours. In this new plot (Fig 5, right), three individual profiles on 3 January 2015 are presented. Differences of up to 15 K are observed between the first profiles (at 03:00 UTC) and the third one (at 13:00 UTC) confirming our previous statement. We have modified the text in the manuscript in order to mention these results in the way that can be read below.

**Author's changes in the manuscript:** p. 9, lines 5-9
"Figure 5 (right) shows an example of strong variation of temperature in the stratosphere for a winter day (3 January 2015). In this case, the temperature changed up to 15 K for some altitudes in the course of only 10 hours. These measurements show the importance of continuous observations for a fixed location, since the important variations in temperature observed cannot be captured by only occasional measurements or measurements with poor temperature resolutions."

**7. Comments from the referee:** Page 11 - It is stated that, above 35km the RS profiles were extrapolated using TEMPERA profiles. But the top altitude of RS profiles varies, so exactly what does this mean? Were only RS profiles which reached 35km included. If there was data above this was it included (instead of using the TEMPERA data)?

**Author's response:**
What we did was to interpolate each individual RS profile to the altitude grid of TEMPERA, and then fill the existing gap between each individual RS profile and TEMPERA profile (RSs usually do not reach altitudes higher than 35) with TEMPERA measurements in order to use the averaging kernels of TEMPERA in the convolution of the RS profiles. In order to clarify this point, we have modified the statement of page 11 in the way that can be below.

**Author's changes in the manuscript:** p. 11, line 6-8
"The RS profiles were interpolated to the altitude grid of TEMPERA radiometer, and completed in the upper part with the TEMPERA measurements, since RSs usually do not reach altitudes higher than 30-35 km. Afterwards, the profiles were convolved using the averaging kernels of TEMPERA."

**8. Comments from the referee:** Figures 7, 9, and 12 – The most piece of information on these is the comparison between TEMPERA and the convolved retrievals from the other instruments. Since this is the case, it would be best to plot the interpolated line first, and then the convolved and TEMPERA lines on top of this. As the plots are currently shown it is sometimes difficult to tell whether the TEMPERA line coincides with the convolved or the interpolated measurement. Alternatively, the interpolated measurement could even be dropped from these plots.

**Author's response:**
According to the referee's suggestion we have updated these three figures (Fig. 7, 9 and 12). In the new plots a better visualisation of the most interested lines (the convolved profiles from the different instruments and from TEMPERA) can be observed.

**9. Comments from the referee:** Figures 7 and 9 – is the black deviation line TEMPERA vs. RS interp or RS conv? In fact, throughout much of the text it is not clear whether convolved or unconvolved RS and/or MLS data is being used.

**Author's response:**
All the comparisons between TEMPERA and the different datasets have been carried out using the convolved profiles of the latter. This point has been indicated for each comparison. However, in order to make clearer for Fig. 7 as the referee suggest we have explicitly indicated it in the description of this figure. The sentence reads as is indicated below.

**Author's changes in the manuscript:** p. 11, line 12
"The temperature deviations along this period between TEMPERA and the convolved measurements from RS are shown in the lower panels (black lines)."

**10. Comments from the referee:** The step in the data shown in Figure 7 very helpful in that it provides a useful measure of the systematic errors in these retrievals. I applaud the investigators for not homogenizing the data between the two periods.

**Author's response:**
We appreciate the positive feedback from the reviewer.

**11. Comments from the referee:** Page 13 – "near time-coincident". Does this mean that the MLS profile was taken during the period of spectral integration for the TEMPERA measurement? If so, please state this. If not, please state what exactly "near time coincident" means. The same applies to the RS comparisons.

**Author's response:**
Yes, as the referee indicated "near time-coincident" in our study means that the MLS and RS profiles were taken during the period of the spectral integration for the TEMPERA measurements. It has been clarified in the manuscript as can be read below.

In the case of RS it was already indicated in the manuscript (page 10, line 20): "The TEMPERA profiles closest in time to the RS launch have been selected in order to do this comparison."

**Author's changes in the manuscript:** p. 13, line 15
"The data were also restricted to cases with near time-coincident between TEMPERA and MLS, which means that the MLS profiles were taken during the period of the spectral integration for the TEMPERA measurements."

**12. Comments from the referee:** Page 15 and Figure 11 – The authors note that: "the bias and the standard deviation observed between MLS and RS is very similar to the values observed in the comparison between TEMPERA and RS in period 2." If do not know, and it seems to be nowhere stated, whether in Figures 8 and 10, the MLS and RS measurements are convolved before comparison with the MWR. If they are (and I think they should be), then the appropriate comparison in Figure 11 would be convolved MLS with convolved RS profiles. This could be added as a dashed line in Figure 11.

**Author's response:**
In Figure 11 we just evaluated the agreement between MLS and RS data in the range where both measurements are comparable. We did not convolve these profiles with the AVK of TEMPERA in order not affect the comparison of these two instruments (RS and MLS) with a third one (TEMPERA). In addition, we have to keep in mind that the measurement response of TEMPERA below 20 km is low, so it would also limit the comparison between MLS and RS in the lower part. In addition, the results after the convolution would be affected by the repair of the attenuator in TEMPERA (since it also affects the AVK). For all these reasons and in order to make our statistics comparable with other studies (e.g., Schwartz et al., 2008) we just compared the profiles from both instruments without convolving them.

**13. Comments from the referee:** Figure 17 – The legend is a bit confusing. Please make the 4 instrument lines solid and thick enough so that one can distinguish lidar and

WACCM. Then separately show two styles of lines, one for Period 1 and one for Period 2.

**Author's response:**
We would like to point out that lidar measurements are only available above 29 km and that in this figure there is only a small range where the mean deviation for both instruments (lidar and WACCM) are almost identical (around 30 km). Below 29 km only the profile from WACCM is shown (blue line) so there is not any overlap with the one corresponding to the lidar (black line). Respect to the styles of the lines we are already representing different styles for both periods as the referee suggests: dashed lines for period 1 and solid line for period 2.

**14. Comments from the referee:** Table 1 – Since Period 1 and Period 2 are presented everywhere else, why is only Period 2 in this table?

**Author's response:**
The idea was to present in this table the values that can characterise better the accuracy and precision of TEMPERA radiometer. Since during the first period TEMPERA was operating with a defective attenuator we decided to show only the values corresponding to period 2 that is when the instrument was operating in optimal conditions.

---

## Author Comment (AC2) · 16 Oct 2017

We are very grateful to referee 2 for his positive feedback of our study.